# After 100 Years of BCG Immunization against Tuberculosis, What Is New and Still Outstanding for This Vaccine?

**DOI:** 10.3390/vaccines10010057

**Published:** 2021-12-31

**Authors:** Mario Alberto Flores-Valdez

**Affiliations:** Biotecnología Médica y Farmacéutica, Centro de Investigación y Asistencia en Tecnología y Diseño del Estado de Jalisco, A.C., Av. Normalistas 800, Col. Colinas de la Normal, Guadalajara 44270, Jalisco, Mexico; floresv@ciatej.mx; Tel.: +52-33-3345-5200 (ext. 1301)

**Keywords:** tuberculosis, BCG, BCGΔBCG1419c, VPM1022, rBCG

## Abstract

In 2021, most of the world was reasonably still concerned about the COVID-19 pandemic, how cases were up and down in different countries, how the vaccination campaigns were ongoing, and most people were familiar with the speed with which vaccines against SARS-Co-V2 were developed, analyzed, and started to be applied in an attempt to curb the pandemic. Because of this, it may have somehow passed relatively inadvertently for people outside of the field that the vaccine used to control tuberculosis (TB), *Mycobacterium bovis* Bacille Calmette-Guérin (BCG), was first applied to humans a century ago. Over these years, BCG has been the vaccine applied to most human beings in the world, despite its known lack of efficacy to fully prevent respiratory TB. Several strategies have been employed in the last 20 years to produce a novel vaccine that would replace, or boost, immunity and protection elicited by BCG. In this work, to avoid potential redundancies with recently published reviews, I only aim to present my current thoughts about some of the latest findings and outstanding questions that I consider worth investigating to help develop a replacement or modified BCG in order to successfully fight TB, based on BCG itself.

## 1. Introduction

In 2021, *Mycobacterium bovis* Bacille Calmette-Guérin (BCG), reached a century since its first application to humans. Quite comprehensive reviews have already been published this year [1,2]. On the other hand, on celebrating the 100 years since BCG’s first administration to humans, experts in the TB field already gave us an account of what specific and unspecific benefits and shortcomings have been discovered in this century of use [3,4]. Therefore, this work does not consider several aspects related to vaccine development, for which the reader may refer to other publications [1,2,4,5]. Also, topics such as innate trained immunity or the effect of BCG on cancer, which were comprehensively discussed by renowned experts in other publications in 2021 [4,6,7,8], are outside the scope of this work. This work also does not deal with sub-unit vaccines, viral vectored, or the use of genetically modified mycobacteria other than BCG, or non-tuberculous mycobacteria to protect against TB. Here, I aim to present my current thoughts about some of the latest findings and outstanding questions that I consider worth investigating to help develop a replacement or modified BCG in order to successfully fight TB, based on BCG itself.

## 2. Benefits and Drawbacks of Current BCG

*Mycobacterium bovis* Bacillus Calmètte-Guèrin (BCG) is the only vaccine currently available and approved by the World Health Organization (WHO) to prevent (reduce the incidence of) TB in humans using a prophylactic immunization strategy. BCG has been used as a vaccine in humans since 1921, and to date more than 3 billion doses have been administered, and nearly 115 million additional doses are applied annually to about 80% of children in the world [9]. BCG protects children against meningeal and miliary TB when administered as a prevention of disease vaccine in non-infected children, its main drawback being that its efficacy widely varies when protection against pulmonary TB is considered. Here, a recent systematic review and meta-analysis leads one to suggest that BCG efficacy depends on timing of exposure to *M. tuberculosis* after vaccination [10].

We should keep in mind that currently, BCG vaccination in humans is performed with more than a single organism, which are produced by different manufacturers around the world [11]. This has prompted speculations of the potential contribution of the known variations among different BCG sub-strains to variable efficacy [12,13], although current evidence is lacking as to whether this is of true clinical relevance. What has been recently demonstrated is that licensed BCG vaccines markedly differ in bacterial viability, RNA content and innate immune activation [14]. Among potential explanations for these variations, manufacturing processes that are difficult to standardize have been suggested. There, production using fermentation and defined chemical media have been shown to affect viability [15]. What other consequences that a change in the method of manufacturing BCG would have in terms of efficacy of protection remain to be determined. 

Another drawback was declared in 2018 by the WHO, which deemed that BCG was not effective as a therapeutic vaccine against post-exposure active and latent tuberculosis infection (LTBI) [16]. The so-called LTBI state has recently been proposed to encompass several different and dynamic interactions between humans and the pathogen, leading the authors to suggest a shift from the use of LTBI to that of “tuberculosis infection-no disease” (TBI-ND) as a way to reflect that, at the end of the day, people in this situation do not shown signs nor symptoms of active TB disease [17]. What the real numbers of people with LTBI/TBI-ND are is a difficult question to answer, and we should be aware that it has been recently proposed that the number of people harboring live *M. tuberculosis* may be substantially lower than the 2 billion previously thought [18]. If we were to arbitrarily assume that only 10% of the population are estimated to harbor LTBI/TBI-ND, this still amounts to 200 million people, which is very close to recent figures of the COVID-19 pandemic (a little over 250 million at the time of writing this work according to https://www.worldometers.info/coronavirus/, accessed on 9 December 2021).

Going by the currently accepted definition of LTBI/TBI-ND and the estimated lifetime risk of reactivation from it, a mathematical modelling estimated that in China, by 2050, up to 75% of new TB cases will be due to reactivation from LTBI/TBI-ND, rather than new infections [19]. In fact, another model that includes data from China, South Africa, and India, suggested that vaccines preventing disease in *M. tuberculosis*-infected populations would have the greatest impact by 2050 (10-year, 70% efficacy against disease, incidence rate reduction 51%, 52%, and 54% in China, South Africa, and India, respectively) [20].

Based on these data, even though a potential replacement for BCG, or a booster to it, that would be applied to non-infected people will surely reduce the burden of TB worldwide, it would be useful to devote resources to incorporate into the current pipeline (https://www.tbvi.eu/what-we-do/pipeline-of-vaccines/, accessed on 9 December 2021), vaccine candidates and relevant models that address the events leading to and regulating exit from, LTBI/TBI-ND.

Here, we can mention that models that could be applied to determine efficacy of protection against reactivation from LTBI/TBI-ND have been developed by coinfection with *M. tuberculosis* and Simian Immunodeficiency Virus (SIV) in rhesus [21,22] and cynomolgus macaques [23,24]. The study by White et al. tested the effect of BCG on reactivation of TB in cynomolgus macaques. There, it was found that the disease burden and pathology was lower in the group that received BCG compared to unvaccinated controls [24]. The authors acknowledged that the number of animals used in their study was limited and, therefore, further studies may be required to validate the model for application in TB vaccine efficacy testing. On the other hand, a recent study comparing active and LTBI/TBI-ND models in rhesus macaques allowed immune cellular populations associated with either active or LTBI-TBI-ND to be dissected [25]. These results could be relevant when determining possible changes in these populations in models leading to reactivation of TB. 

To the best of my knowledge, only VPM1002 [26] and BCGΔBCG1419c [27] have shown improved efficacy compared with their parental BCG strains in murine models of reactivation from subclinical infection, which aim to resemble LTBI/TBI-ND. Of note, the models used in each study were different and with the known limitations of mouse models of TB in resembling only some aspects but not all occurring during human disease. It would be interesting to determine whether any of these rBCGs (or any other current TB vaccine candidate) is able to control better than TB the reactivation from LTBI/TBI-ND reported in the non-human primate models just described above.

## 3. Latest Findings Regarding BCG Efficacy in Clinical Trials and in Advanced Preclinical Models

In the fight against TB, BCG has not been conclusively ruled out, as it has recently been shown that the intradermal revaccination with BCG to humans with LTBI (defined as a positive Quantiferon^®^-QFT- result with no signs nor symptoms of active disease) can reduce the rate of QFT^®^ conversion, which might result from the capacity to either reduce reactivation from LTBI/TBI-ND or to reduce new infections [28].

On the other hand, because it is currently accepted that TB is transmitted via inhalation of infectious droplets, delivery of BCG (or other novel modified BCGs) via the aerosol, intranasal or intratracheal routes has gained momentum in recent years, given that they improved immunogenicity and/or protection in small animal models as well as nonhuman primates compared with subcutaneous (s.c.) or intradermal (i.d.) immunization [29,30,31,32,33,34]. However, it is worth noting that in rhesus macaques, no difference was found in the survival of animals vaccinated with BCG via i.d. or aerosol, against ultralow dose challenge with *M. tuberculosis* Erdman strain (estimated in 4 colony-forming units (CFU)) [35]. In this work, i.d. and aerosol routes resulted in similar total pathology score, with i.d. vaccination significantly reducing lung score compared with unvaccinated controls, whereas aerosol delivery outperformed i.d. immunization in reducing extrathoracic injury scores [35]. Does this imply that i.d. vaccination would still confer potential advantages over aerosol delivery of BCG to control pulmonary TB?

Despite these encouraging results for the utilization of vaccination routes that emulated the natural route of infection (aerosol), quite interestingly it was intravenous vaccination with BCG that surpassed the benefits of aerosol or intradermal vaccination of rhesus macaques [36,37], when animals were challenged with low doses of *M. tuberculosis* Erdman (4–36 CFU in [36], and 100 CFU in [37]). Furthermore, it has just been shown that rhesus macaques that received BCG intravenously (i.v.), had superior IgM antibody responses in the plasma and the lungs compared to traditional intradermal BCG administration, and correlated with reduced *M. tuberculosis* burdens [38]. I wonder whether any of the current rBCGs in development would protect better than BCG if they were applied by a route other than the subcutaneous/intradermal.

Like humans, nonhuman primates differ in their response to BCG and susceptibility to TB. Here, while cynomolgus macaques were well protected by BCG, rhesus macaques were not [39] upon a high dose (3000 CFU) challenge with *M. tuberculosis* Erdman [39]. Considering the improved control on TB pathogenesis that i.v. vaccination with BCG confers to rhesus monkeys, we can raise some questions that might be worth providing experimental evidence about, such as: (a) would i.v. vaccination protect cynomolgus macaques when employed at a lower dose compared to rhesus macaques? (b) would i.v. vaccination of rhesus monkeys protects against challenge with higher *M. tuberculosis* doses, so that they might be as protected as cynomolgus macaques? (c) Would i.v. vaccination with BCG protect against reactivation from LTBI-TBI-ND? (d) Would i.v. (or any other route of administration) of BCG be efficacious against multidrug-resistance (MDR) *M. tuberculosis* strains by themselves or with adjunct antibiotic therapy? (e) How effective and translationally relevant would these non-human primate models be should they be infected with *M. tuberculosis* other than the commonly used H37Rv and Erdman strains? (f) Would a change in BCG manufacturing practices affect efficacy of protection observed thus far in these models?

Each study that explored novel vaccination routes has undoubtedly called to attention the possibility of translating it into the vaccination of humans against TB. Notwithstanding their potential benefits, it should be considered that the use of routes other than intradermal vaccination might pose ethical, logistic, or regulatory issues that would need to be resolved before a mass utilization for said novel route could be established. The main points that should be considered in future research regarding BCG, in our vision, are summarized in Figure 1.

## 4. Novel Live, Attenuated, BCG-Based Vaccine Candidates: What Has Been Shown and What Is Still Missing?

Two main strategies have been followed to produce modified, recombinant BCG strains that aim to improve its efficacy against TB: deleting genes from the BCG genome (knockouts) and inserting mycobacterial genes into the BCG genome (knock-ins, KI). I will briefly describe a few examples from each category below.

rBCGs based on KO: (1) VPM1002 (BCGΔureC::llo, that is, devoid of the urease-encoding *ureC* gene and with an insertion of the *L. monocytogenes* listeriolysin gene *llo*) is the most advanced vaccine candidate based on BCG Danish, as it has satisfactorily completed phases I and II of clinical trials, showing safety and immunogenicity profiles comparable to those of current BCG [40,41], and a phase III study is currently in progress (ClinicalTrials.gov Identifier: NCT04351685). (2) BCGΔureC::hlyΔnuoG (a VPM1002-related variant with deletion of the *nuoG* gene), which led to reduced replication in lungs and spleens of *M. tuberculosis* H37Rv or W/Beijing-infected mice compared with wild-type BCG Danish [42], especially during chronic TB. I could not find a report of what would be the effect of deleting *nuoG* from the actual VPM1002 in its efficacy against TB. (3) BCGΔzmp1 (devoid of the zinc metalloprotease encoding gene *zmp1*), which improved protection against TB in the highly susceptible guinea pig model [43], and is awaiting further reports of its development, regardless of its being based on BCG Danish or Pasteur. (4) BCGΔBCG1419c (based on BCG Pasteur and devoid of the c-di-GMP phosphodiesterase encoding gene *BCG1419c*), which in its first-generation version (based on BCG Pasteur 1173P2, bearing a hygromycin resistance gene instead of *BCG1419c*) has shown improved efficacy over its parental BCG in reducing *M. tuberculosis* H37Rv replication in lungs of (a) BALB/c mice with chronic TB, (b) while also reducing H37Rv loads in lungs and pathology in a model of reactivation from LTBI/TBI-ND in B6D2F1 mice [27]. It also (c) reduced lung pathology during chronic H37Rv infection of C57/BL/6 mice [44]. A more recently developed, second-generation version of BCGΔBCG1419c (devoid of antibiotic resistance markers and based on BCG Pasteur ATCC 35734 strain) improved protection against pulmonary and extrapulmonary TB in guinea pigs infected with H37Rv [45]. We continue developing BCGΔBCG1419c because despite differences in our experimental designs and models employed, the common feature we observed in mice and guinea pigs is the improvement in reducing pulmonary damage. In this regard, quoting the work of Dannenberg and Collins “Progressive pulmonary tuberculosis is not due to increasing numbers of viable bacilli in rabbits, mice, guinea pigs, and humans who develop paucibacillary disease, but is due to a continuous host response to mycobacterial products” [46], therefore, this could be a special benefit of BCGΔBCG1419c over other vaccine candidates, which remains to be formally tested.

rBCGs based on KI: (1) rBCG30, resulting from the overexpression of the 30KDa antigen encoded by the *Ag85b* gene, which improved protection of vaccinated guinea pigs compared with parental BCG [47]. rBCG30 was immunogenic in human volunteers [48] but despite this I could not find further reports of its clinical evaluation. (2) BCG::RD1, produced upon introduction of the region of difference 1 (RD1, a genome fragment present in virulent *M. tuberculosis* and absent from BCG that includes among other potent immunogens those encoded by the *esat6* and *cpf10* genes) into BCG Pasteur 1173P2, which was more effective in reducing TB in C57BL/6 mice and guinea pigs [49] but was disregarded for further development because of its increased virulence compared with BCG [50]. Because of this, (3) BCG::RD1^Mmar^ (BCG containing RD1 from *M. marinum*) was developed, and found to significantly reduce *M. tuberculosis* HN878 and M2 loads in the lungs and spleens of C57BL/6 mice compared with parental BCG, but with no significant decrease of pulmonary inflammation [51]. Whether BCG::RD1^Mmar^ will continue in development remains to be seen. (4) Another refinement to the use of RD1 was recently reported in the BCG::ESAT6-PE25SS strain, which secretes full-length ESAT-6 via the ESX-5 secretion system, and reduced *M. tuberculosis* H37Rv loads in lungs as well lung pathology in intratracheally-vaccinated C57BL/6 more than parental BCG [52], so it will be interesting to see further development of this vaccine candidate. (5) BCG::phoPR, an interesting candidate based on the fact that the *phoPR* genes from BCG Pasteur were introduced into BCG Japan to restore it from its natural point mutations, and which prolonged survival of guinea pigs infected with *M. tuberculosis* H37Rv compared with parental BCG [53]. Whether this specific recombinant BCG strain will undergo further development or will be the basis of possible restoration of gene function in other rBCG vaccine candidates remains to be seen.

Considering that most novel rBCGs have been tested by using s.c. vaccination routes and their efficacy evaluated in models of active TB, it is worth asking the following questions: (a) would i.v. vaccination result in improved protection compared to other routes for all, some, or none of these rBCGs? (b) Would vaccination by the i.v. (or any other) route protect against reactivation from LTBI-TBI-ND equally well or more than current BCG? (c) Would any of these rBCGs be efficacious against multidrug-resistance (MDR) *M. tuberculosis* strains by themselves or with adjunct antibiotic therapy? (d) How effective would these rBCGs be against TB caused by *M. tuberculosis* other than the commonly used H37Rv and Erdman strains? (e) Would modifications in manufacturing practices affect efficacy of protection conferred by rBCGs?

## 5. The Need of Correlates of Protection

It is worth noting that there is no current definite correlate of protection to evaluate vaccine efficacy more precisely, either at the preclinical or clinical settings. This could change in the short term, as omics technologies have helped predict the risk of disease development even several weeks or months in advance [54,55,56,57], and in fact approaches of this type were already suggested to play a relevant role for vaccine development [58]. Along these lines, recent methods to analyze high dimensional complex data, such as t-distributed stochastic neighbor embedding (tSNE), CITRUS (cluster identification, characterization, and regression), COMPASS (a computational framework for unbiased polyfunctionality analysis of antigen-specific T-cell subsets) and mixture models for single-cell assays (MIMOSA) were recently applied to determine the immune response of human peripheral blood mononuclear cells (PBMCs) upon vaccination with either BCG or H4:IC31 at baseline and day 70 post-vaccination [59]. When combined with readouts of infection and its clearance, these might also help shed light on what an effective immune response is and what is entailed to curb the progression of infection (or exit from LTBI/TBI-ND) to disease.

In summary, we can see that despite its century-old utilization to reduce TB worldwide, BCG has recently taught us several lessons, and it continues to be the gold standard for the efficacy of protection, despite the limitations discussed above and elsewhere. Furthermore, BCG could be the basis for a novel, improved, more efficacious and much needed vaccine that reduces the global burden of TB in the next few years.

## Figures and Tables

**Figure 1 vaccines-10-00057-f001:**
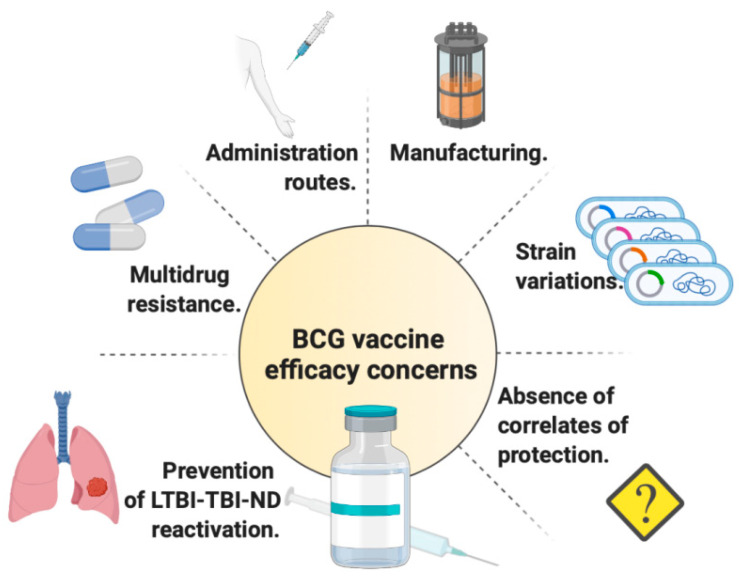
The efficacy of *Mycobacterium bovis* Bacille Calmette-Guérin (BCG) as a vaccine against tuberculosis (TB), and the most recent findings and questions that need to be addressed are summarized here, as potential areas of research for novel TB vaccine candidates. The value and potential of alternate routes of BCG vaccination other than intradermal, in humans, must be explored. How effective TB vaccines are against multidrug-resistant (MDR)-TB should be determined. Novel preclinical models should shed light on the efficacy of BCG and other TB vaccine candidates in preventing reactivation from latent tuberculosis infection (LTBI)/tuberculosis infection-no disease (TBI-ND). Manufacturing processes that are easier to perform and replicate should reduce variations in viability and immunogenicity. Whether BCG strain variation truly impacts on protection to humans or not should be clarified. Correlates of protection are needed and start to emerge via omics tools.

## Data Availability

Not applicable.

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
