# Peer review of "After 100 Years of BCG Immunization against Tuberculosis, What Is New and Still Outstanding for This Vaccine?"

_vaccines, 2021, doi:10.3390/vaccines10010057_

Round 1
Reviewer 1 Report
In the submitted paper by Flores-Valdez, the author brings to our attention some important aspects of BCG vaccination, current trends, and issues, and discusses how the vaccine can be improved. However, the author did not comprehensively address the topic, and did not do justice to the literature. There have been many prominent scientists working on TB/BCG vaccination improvements, for decades, finding and reporting alternative vaccine strategies.
Most importantly, the most dangerous aspect of TB is the emergence of multi-drug resistant (MDR) strains. The effect of the vaccine on MDRs was not discussed at all. The author should address this, and discuss whether vaccines being developed include protection against MDR strains.
Page 4, line 199 Expand IAVI
Author Response
In the submitted paper by Flores-Valdez, the author brings to our attention some important aspects of BCG vaccination, current trends, and issues, and discusses how the vaccine can be improved. However, the author did not comprehensively address the topic, and did not do justice to the literature. There have been many prominent scientists working on TB/BCG vaccination improvements, for decades, finding and reporting alternative vaccine strategies.
R: I understand and sincerely appreciate this concern. I must say that this work was originally submitted as an “Opinion” type of article, and the editorial team changed it to become a “Review” and therefore you found a number of limitations in covering the literature. I still did not endeavor to present a comprehensive review, given that many of them were published this year and contain fundamental information that I do not want to repeat. The contribution of my work would be to highlight some potential avenues for further research of BCG and rBCGs. I hope that this revised version successfully achieves this aim.
Most importantly, the most dangerous aspect of TB is the emergence of multi-drug resistant (MDR) strains. The effect of the vaccine on MDRs was not discussed at all. The author should address this and discuss whether vaccines being developed include protection against MDR strains.
R: This is a very thoughtful suggestion and I sincerely appreciate it. To the best of my knowledge, no rBCG has been tested against MDR strains. As a way of covering this topic, I included a few lines regarding this “be efficacious against multidrug-resistance (MDR) M. tuberculosis strains by themselves or with adjunct antibiotic therapy?”.
Page 4, line 199 Expand IAVI
In the submitted paper by Flores-Valdez, the author brings to our attention some important aspects of BCG vaccination, current trends, and issues, and discusses how the vaccine can be improved. However, the author did not comprehensively address the topic, and did not do justice to the literature. There have been many prominent scientists working on TB/BCG vaccination improvements, for decades, finding and reporting alternative vaccine strategies.
R: I understand and sincerely appreciate this concern. I must say that this work was originally submitted as an “Opinion” type of article, and the editorial team changed it to become a “Review” and therefore you found a number of limitations in covering the literature. I still did not endeavor to present a comprehensive review, given that many of them were published this year and contain fundamental information that I do not want to repeat. The contribution of my work would be to highlight some potential avenues for further research of BCG and rBCGs. I hope that this revised version successfully achieves this aim.
Most importantly, the most dangerous aspect of TB is the emergence of multi-drug resistant (MDR) strains. The effect of the vaccine on MDRs was not discussed at all. The author should address this and discuss whether vaccines being developed include protection against MDR strains.
R: This is a very thoughtful suggestion and I sincerely appreciate it. To the best of my knowledge, no rBCG has been tested against MDR strains. As a way of covering this topic, I included a few lines regarding this “be efficacious against multidrug-resistance (MDR) M. tuberculosis strains by themselves or with adjunct antibiotic therapy?”.
Page 4, line 199 Expand IAVI
R: Following concerns about the potential bias of this work in citing my own research, I removed the last paragraph regarding BCGDBCG1419c development and therefore removed the text about IAVI.
R: Following concerns about the potential bias of this work in citing my own research, I removed the last paragraph regarding BCGDBCG1419c development and therefore removed the text about IAVI.
Reviewer 2 Report
Flores-Valdez presents a short review on the BCG vaccine with an eye toward development of a better BCG. The review as is suffers from two main ways. First, the review in its present form is not accessible to readers outside of the BCG field and must be enhanced with a stronger introduction to BCG itself with a way to track signs of efficacy. Figures or tables to help the reader understand the points are important and missing. Second, the author needs to distance himself from the review. While using several disclaimers about his work, the author inadvertently makes this an article about his work and its significance. I would strongly suggest the author either completely remove reference to their work or diminish the amount of time discussed. Without improving the connection to the greater audience and diminishing the connection to the author's work, this paper cannot yet be published.
Author Response
Flores-Valdez presents a short review on the BCG vaccine with an eye toward development of a better BCG. The review as is suffers from two main ways. First, the review in its present form is not accessible to readers outside of the BCG field and must be enhanced with a stronger introduction to BCG itself with a way to track signs of efficacy. Figures or tables to help the reader understand the points are important and missing.
R: I understand this situation and as for BCG pros and cons, I updated the revised version with an aim to more clearly present these. As this is a work intended for those experienced with BCG, I rather than using figures or tables I went directly to point out some questions I consider relevant for this field.
Second, the author needs to distance himself from the review. While using several disclaimers about his work, the author inadvertently makes this an article about his work and its significance. I would strongly suggest the author either completely remove reference to their work or diminish the amount of time discussed. Without improving the connection to the greater audience and diminishing the connection to the author's work, this paper cannot yet be published.
R: I understand and sincerely appreciate this concern. I must say that this work was originally submitted as an “Opinion” type of article, and the editorial team changed it to become a “Review” and therefore you found a number of limitations in covering the literature. I still did not endeavor to present a comprehensive review, given that many of them were published this year and contain fundamental information that I do not want to repeat. The contribution of my work would be to highlight some potential avenues for further research of BCG and rBCGs. I hope that this revised version successfully achieves this aim. Moreover, following concerns about the potential bias of this work in citing my own research, I removed the last paragraph regarding BCGDBCG1419c development.
Round 2
Reviewer 2 Report
The author has responded positively to all points except changing the review article to something that is accessibly with visuals. I did a brief search through the journal and all reviews I ran across, even those that were focused had some visual component (e.g. figures or tables). I don't think the idea that this is supposed to be for a BCG expert and so "rather than using figures or tables I went directly to point out some questions I consider relevant for this field." is a useful discussion on why this article, as a review article, needs to be inaccessible and easily connected to others. As it is, the article is a long document and needs something to help visualize the ideas.
Author Response
I sincerely appreciate the opportunity to provide a brief summary of main points as Table 1 and Table 2. I would deeply appreciate the reviewer understanding that this work was not originally intended to be a Review, but rather an Opinion or Perspective type of manuscript. I did not produce tables or figures for other content to avoid being repetitive. Hopefully you will agree with me that the tables provided would be a concise way to convey the main advantages and opportunities in the field of BCG.
Best wishes,
Mario A. Flores-Valdez, Ph.D.